

# A New Hot-Stage Microscopy Technique for Measuring Temperature-Dependent Viscosities of Aerosol Particles and its Application to Farnesene Secondary Organic Aerosol

Kristian J. Kiland[1], Kevin L. Marroquin[1], Natalie R. Smith[2], Shaun Xu[1], Sergey A. Nizkorodov[2], and Allan K. Bertram[1]

[1]Department of Chemistry, The University of British Columbia, Vancouver, British Columbia V6T 1Z1, Canada

[2]Department of Chemistry, University of California Irvine, Irvine, California 92697, USA

*Correspondence to:* Allan K. Bertram (bertram@chem.ubc.ca)

**Abstract.** The viscosity of secondary organic aerosols (SOA) is needed to predict their influence on air quality, climate, and atmospheric chemistry. Many techniques have been developed to measure the viscosity of micrometer-sized materials at room temperature, however, few techniques are able to measure viscosity as a function of temperature for these small sample sizes. SOA in the troposphere experience a wide range of temperatures, so measurement of viscosity as a function of temperature is needed. To address this need, a new method was developed based on hot-stage microscopy combined with fluid dynamics simulations. To validate our technique, the viscosity of 1,3,5-tris(1-naphthyl)benzene was measured and compared with values reported in the literature. Good agreement was found between our measurements and literature data. As an application to SOA, the viscosity as a function of temperature for lab-generated farnesene SOA was measured, giving values ranging from $3.4 \times 10^6$ Pa s at 51 °C to $2.6 \times 10^4$ Pa s at 67 °C. These values were significantly higher (1–2 orders of magnitude) than values predicted using a parameterization from DeRieux et al. (2018), the chemical composition of the SOA measured with a high-resolution mass spectrometer, and assuming a fragility of 10 (as done previously). These results illustrate that our new experimental approach provides important data for testing methods used for predicting the viscosities of SOA in the atmosphere.





## 1 Introduction

Secondary organic aerosols (SOA) are formed in the atmosphere by the oxidation of volatile and intermediate volatile organic compounds followed by gas-to-particle partitioning of oxidation products (Hallquist et al., 2009; Jimenez et al., 2009). SOA can also be formed by condensed-phase reactions (Ervens et al., 2011; McNeill, 2015). SOA is a major contributor to respirable particulate matter in urban environments, which leads to negative health effects (Lelieveld et al., 2015; Nel, 2005; Pope and Dockery, 2006; Shiraiwa et al., 2017a). SOA also influences climate directly by scattering and absorbing incoming solar radiation, as well as indirectly by acting as cloud condensation nuclei and ice nucleating particles (Masson-Delmotte et al., 2021; Myhre et al., 2013).

Recent work has shown that predictions of the size and mass of SOA can be sensitive to diffusion rates in SOA (Han et al., 2019; Kim et al., 2019; Shiraiwa et al., 2011, 2013; Shiraiwa and Seinfeld, 2012; Vander Wall et al., 2020; Ye et al., 2018; Yli-Juuti et al., 2017; Zaveri et al., 2014, 2018, 2022; Zhang et al., 2019a). Diffusion rates within SOA influence heterogeneous chemistry and condensed-phase photochemistry (Alpert et al., 2021; Dalton and Nizkorodov, 2021; Fitzgerald et al., 2016; Gržinić et al., 2015; Li and Knopf, 2021; Li et al., 2021; Marshall et al., 2016; Schmedding et al., 2020; Shiraiwa et al., 2011; Steimer et al., 2014) and the long-range transport of pollutants (Friedman et al., 2014; Keyte et al., 2013; Mu et al., 2018; Shrivastava et al., 2017; Zelenyuk et al., 2012). Water uptake, and hence ice nucleating properties of SOA, depend on diffusion rates (Fowler et al., 2020; Ignatius et al., 2016; Ladino et al., 2014; Lata et al., 2021; Lee et al., 2020; Murray et al., 2010; Tumminello et al., 2021). Diffusion rates can also influence crystallization rates within SOA-containing particles (Bodsworth et al., 2010; Fard et al., 2017; Ji et al., 2017; Murray, 2008). As a result, measurements of diffusion rates within SOA as a function of key atmospheric variables, namely relative humidity (RH) and temperature (Porter et al., 2021), are needed. Alternatively, measuring the viscosity of SOA as a function of RH and temperature can be used together with the Stokes-Einstein relation or the fractional Stokes-Einstein relation to calculate diffusion rates within SOA (Evoy et al., 2019, 2020, 2021; Ingram et al., 2021; Price et al., 2016).

Measurement of the viscosity of SOA is challenging due to the exceedingly small amount of SOA material (typically in the microgram range) that can be collected from the atmosphere or environmental chambers in a reasonable amount of time. Nevertheless, several techniques have been developed to measure the viscosity of SOA as a function of relative humidity (RH) at ambient temperatures (Reid et al., 2018). In addition, atomic force microscopy (AFM) techniques have recently been developed for probing the RH-dependent phase state of aerosols (Lee et al., 2020; Lee and Tivanski, 2021; Madawala et al., 2021). Although some of these methods could be extended to different temperatures by making modifications to the techniques (e.g. Logozzo and Preston (2022)), only three methods have been developed and used to measure the viscosity of SOA as a function of temperature. Rothfuss and Petters used a dimer relaxation technique to measure viscosities of SOA in the range of $5 \times 10^5$ to $2 \times 10^7$ Pa s at temperatures ranging from −15 to 80 °C (Rothfuss and Petters, 2016). This technique has the advantage that the viscosity of the SOA is determined without collecting the material on substrates. On the other hand, this means samples cannot be collected and measured at a different time and different location using this technique. Jarvinen et al. (2016) used light scatter techniques to measure viscosities of SOA at ~$10^7$ Pa s and for temperatures between −38 and 10 °C (Jarvinen





et al., 2016). Similar to the dimer relaxation technique, the viscosity of the SOA is determined without collecting the material. Qin et al. (2021) coupled AFM to a temperature-controlled sample module to measure the viscosities in the range of ~$10^{-3}$ to $10^{-1}$ Pa s at temperatures between 15 and 95 °C for organic particles collected on substrates (Qin et al., 2021). Although SOA in the atmosphere is frequently > $10^{-1}$ Pa s (Maclean et al., 2021a; Shiraiwa et al., 2017b),

at this point, the temperature-dependent AFM technique has only been used to probe particles with viscosities ≤ $10^{-1}$ Pa s.

Hot-stage microscopy has been used previously to measure the temperature-dependent viscosity of relatively large amounts of samples (≳ 70 mg needed for sample preparation) (Garcia-Valles et al., 2013; Pascual et al., 2001, 2005; Scholze, 1962). In this method, the material of interest, in the form of a powder, is pressed into a cube or cylinder

with length scale on the order of ~3000 μm. The cube or cylinder is then heated at a fixed rate and viewed with an optical microscope. From images recorded during heating, several characteristic temperatures are determined including the temperature of first shrinkage, the temperature of maximum shrinkage, and the temperature for half-ball formation. Viscosities are then estimated for these characteristic temperatures by comparison with characteristic temperatures of laboratory standards having well-known viscosities. For this procedure to be accurate, the material of

interest must have a similar surface tension, contact angle, and slip length (a measure of resistance to flow at a solid substrate) as the laboratory standards. Note that this procedure will not work with SOA due to the small amount of SOA material that can be collected on a reasonable time scale. In addition, SOA can have a wide range of properties (and hence surface tensions, contact angles, and slip lengths), so what to use as an appropriate laboratory standard is unclear.

In the following, we develop a new methodology for measuring the viscosity of SOA as a function of temperature, building on the hot-stage microscopy (HSM) technique discussed above. In our method, many SOA particles are coalesced into one larger particle with an area-equivalent diameter of ~60 to 190 μm , and placed on a hot-stage coupled to an optical microscope. The temperature of the hot-stage is then increased rapidly to a temperature of interest and held at this temperature for an extended time. Images of the particles are recorded as a function of time at this

fixed temperature. If the temperature is hot enough, the shape of the particles changes to reduce the overall surface energy of the system. Fluid dynamic simulations are then used to determine the viscosity of the material from the change in the shape of the particles during the observation time at the fixed temperature. The fluid dynamics simulations explicitly account for the surface tension, contact angle, and slip length of the material. The technique can be used to measure the viscosity of small amounts of SOA (< 1 mg of material is needed for sample preparation). The

viscosity measurements described here can be made on SOA collected on substrates, and as result, they do not have to be made at the same location or time as SOA generation or collection. The methodology presented here is complementary to the dimer relaxation technique and light scattering technique discussed above for measuring temperature-dependent viscosities of SOA. Our new methodology also uses inexpensive equipment that is available in many laboratories.

To validate our HSM technique, the viscosity of 1,3,5-tris(1-naphthyl)benzene (TαNB) (Fig. 1a) was measured as a function of temperature and the results were compared to literature values. TαNB was chosen for the validation

measurements since this material readily forms an amorphous solid at room temperature (a prerequisite for our measurements) and the temperature-dependent viscosity of this material has been reported in the literature (Plazek et al., 1999). We show that the viscosities determined with our new methodology are consistent with the viscosities reported in the literature for TαNB.


As the first application of our new methodology, the temperature-dependent viscosity of SOA generated from the photooxidation of farnesene was measured. Farnesene (Fig. 1b), an acyclic sesquiterpene, can be a significant component of biogenic volatile organic compounds (VOCs) emitted to the atmosphere from trees, shrubs, grasslands, and crops (Bouvier-Brown et al., 2009; Geron and Arnts, 2010; Helmig et al., 2006; Li and Xie, 2014; Ylisirniö et al.,


2020). Plants have increased emission rates of farnesene and other sesquiterpenes under periods of insect-herbivory, heat, and drought stress, which is projected to increase due to climate change, making farnesene an interesting VOC to study (Faiola et al., 2019; Mentel et al., 2013). Previous work has shown that farnesene reacts with atmospheric oxidants such as ozone ($O_3$), hydroxyl radicals (OH), and nitrate radicals ($NO_3$) to form SOA with a high yield (Jaoui et al., 2013). Acyclic terpenes, such as farnesene, tend to break down into smaller products during $O_3$ oxidation but


not OH or $NO_3$ oxidation leading on complex effects on SOA yields from plants depending on the prevailing oxidant (Faiola et al., 2019). The viscosity of SOA generated from a mixture of VOCs including farnesene has been studied at room temperature (Smith et al., 2021), but the viscosity of SOA generated from only farnesene has not been studied at room temperature or as a function of temperature.



**Figure 1** Chemical structures of **(a)** 1,3,5-tris(1-naphthyl)benzene and **(b)** farnesene.

In addition to measuring the temperature-dependent viscosities of farnesene SOA, we also determined the chemical composition of the farnesene SOA using high-resolution mass spectrometry. The measured viscosity was then compared with predictions based on the chemical composition and a parameterization presented in DeRieux et al.



(2018). The DeRieux et al. (2018) parameterization has often been used to predict the viscosity and glass transition temperatures of SOA (DeRieux et al., 2018; Ditto et al., 2019; Gervasi et al., 2020; Maclean et al., 2021b; Pratap et al., 2018; Riva et al., 2019; Schum et al., 2018; Slade et al., 2019; Tikkanen et al., 2020; Wolf et al., 2019; Zhang et al., 2019a, 2019b), but the accuracy of this parameterization for predicting the viscosity of SOA has not been previously tested for SOA produced from acyclic terpenes.

## 2 Materials and methods

### 2.1 Generation of amorphous solid particles

For these experiments, amorphous solid particles (i.e. non-crystalline particles that do not flow significantly on a timescale of hours) with dimensions ~20 – 300 μm were needed. Below is a description of how we generated amorphous solid particles consisting of TαNB and farnesene SOA.

To generate amorphous solid particles of TαNB, crystals of 1,3,5-tris(1-naphthyl)benzene (TRC Canada) were placed on a glass slide, and the temperature of the slide was increased to a temperature above the melting point of TαNB (mp = 184 – 186°C). After the crystals had melted, the temperature was rapidly decreased (~2°C s$^{-1}$) to room temperature, resulting in the formation of a glassy material (Magill and Plazek, 1967; Plazek and Magill, 1966). The glassy material was then scraped off the slide with a razor blade, forming amorphous solid particles with area-equivalent diameters

ranging from ~60 to 190 μm. The formation of a glass state was verified by showing that the material flowed at a temperature significantly warmer than the melting temperature (see below).

Farnesene SOA particles were generated by photooxidation of farnesene in a 5 m$^3$ environmental chamber operated in batch mode at 50% RH and room temperature (21-23 °C), similar to previous work (Smith et al., 2021). A farnesene isomer mixture (Sigma-Aldrich, Product#: W383902) was injected into the chamber (8 μl), and a proton-transfer-

reaction time-of-flight mass spectrometer (PTR-ToF-MS; Ionicon model 8000) with $H_3O^+$ as the reagent ion was used to confirm the completeness of injection of the farnesene precursor into the chamber by monitoring m/z 205 (protonated farnesene). The initial mixing ratio of farnesene was ~160 ppb for all experiments. Next, $H_2O_2$ (45 μl of 30 % wt $H_2O_2$ in water) was added to the chamber through a heated inlet (50 °C), resulting in 2 ppm of $H_2O_2$ in the chamber. Photooxidation was initiated by turning on a bank of UV-B lights, which led to the production of OH radicals

by photodissociation of $H_2O_2$ (Smith et al., 2021).

No seeds were used during SOA generation to avoid interference with viscosity measurements. Particle size and number concentration were measured using a scanning mobility particle sizer (SMPS; TSI differential mobility analyzer model 3080 and CPC model 3775). During photooxidation, the mass concentration of SOA in the chamber reached approximately 500 μg m$^{-3}$. After SOA was generated, the UV-B lights were turned off, and SOA particles

were collected onto hydrophobic glass substrates placed on stage 9 of a non-rotating microorifice uniform deposit impactor (MOUDI) operated at 30 L min$^{-1}$ with all the remaining MOUDI stages removed as discussed elsewhere (Smith et al., 2021). Hydrophobic glass slides were generated by coating plain glass slides with FluoroPel 800 (Cytonix USA). Samples were collected for 3 hours. This method of collection resulted in droplets with spherical cap


geometries and diameters of ~30 to 250 μm, formed by the aggregation of smaller SOA particles on the hydrophobic glass substrates. After collection, samples were placed in protective plastic enclosures, sealed with a vacuum food sealer, and stored in a freezer at −20 ℃ until analysis (except for 24 h transit time when the samples had to be shipped to other participating laboratories on ice at 0 ℃). To prepare the sample for the viscosity measurements, some of the SOA material was scraped off the slide using a razor, resulting in amorphous solid particles of the SOA with area-equivalent diameters of ~80 to 170 μm.

## 2.2 Hot-stage microscope

After the amorphous solid particles were generated, one or more of the particles was attached to the flat end of an ultra-fine tungsten needle (Roboz Surgical Instruments Co.) by bringing the end in contact with the amorphous particles. This needle was then placed in the HSM apparatus for the viscosity measurements. The HSM apparatus consists of a temperature control stage (HC321Gi, *INSTEC*) mounted above an optical transmission microscope (Zeiss Axio Observer) (Fig. 2a). The needle was held in place using an aluminum needle holder (Fig. 2b), and the needle was situated so that the flat end of the needle hung over the edge of the viewing window for imaging. This configuration enabled a side-view of the solid amorphous particles during heating.

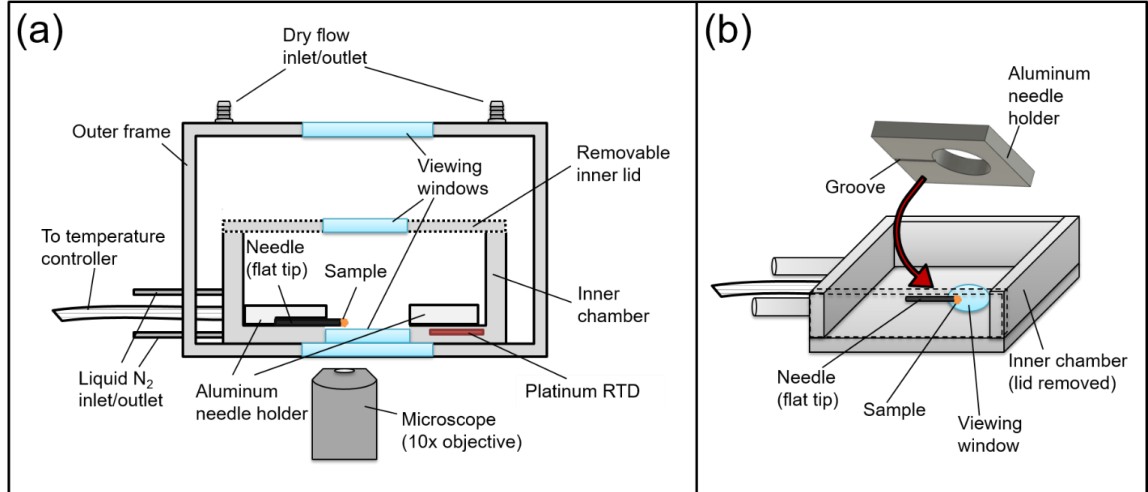

**Figure 2** Schematic of the hot-stage microscopy apparatus, where **(a)** is a cross-section of the whole apparatus and **(b)** shows a perspective view of the inner chamber of the temperature control stage.

The temperature of the hot-stage was controlled by offsetting hot and cold inputs. The cold input was supplied by a liquid nitrogen flow around the walls of the inner chamber and the hot input was supplied by electrical heaters connected to a temperature controller box (mk2000, *INSTEC*). The temperature of the hot-stage was measured with a platinum RTD (Fig. 2a). The temperature offset between the RTD reading and the flat end of the needle was



determined by measuring the melting point of a range of substances (diphenyl ether, 1-octadecanol, glutaric acid, pimelic acid, vanillyl mandelic acid, and cholesterol) placed on the flat end of the needle and by comparing the measurements with melting temperatures reported in the literature for these substances. The temperature ramp used when measuring the melting points was 0.1 °C min$^{-1}$, and images from the melting point experiments were analyzed

with the Zen Microscopy software (Zeiss). From the melting point experiments, a calibration curve was generated (Fig. S1). This calibration curve was used to determine the temperature offset in the viscosity measurements.

## 2.3 Viscosity measurements

In a typical experiment used to determine the viscosity of particles, the temperature control stage was first purged with nitrogen. For the farnesene SOA experiments, the samples were purged with nitrogen for at least 75 minutes before

heating, since farnesene SOA is hygroscopic. The viscosity results were not strongly sensitive to conditioning times between 75 and 200 minutes (Fig. S2). If the farnesene SOA did not reach equilibrium after purging (i.e. they still contained some water), the viscosities reported here should be considered as lower limits to the viscosity of dry farnesene SOA, since water acts as a plasticizer (i.e. decreases the viscosity of highly viscous material). After purging the cell with nitrogen, the temperature of the cell was increased at a rate of ~2 °C s$^{-1}$ to an experimental temperature,

$T_{exp}$. During heating, images were recorded as a function of time. Due to the fast ramp rate, the temperature slightly overshot the desired $T_{exp}$ in each experiment and then oscillated around $T_{exp}$, with the oscillations decreasing with time (e.g. Fig. 3). The start of the experiment (i.e. $t = 0$) was defined as the first image captured after the temperature oscillation was < 2 °C.

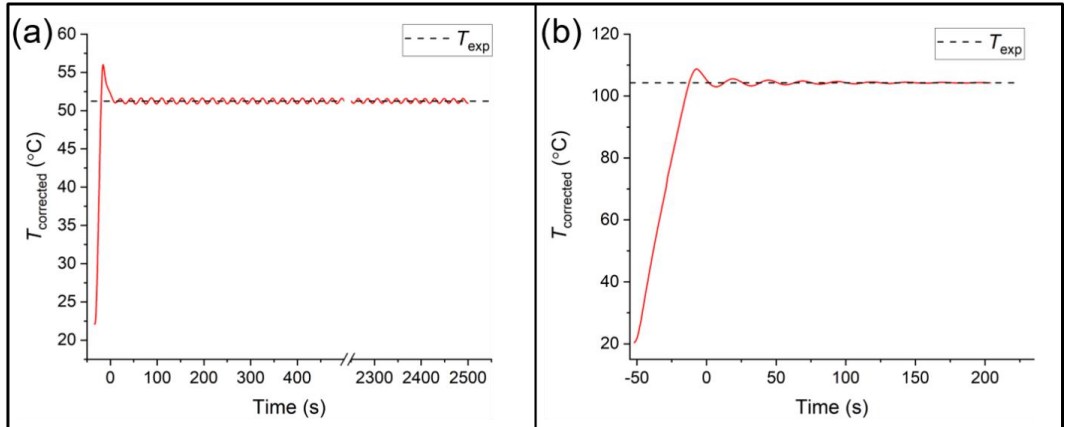

**Figure 3** Temperature profiles from hot-stage microscopy experiments. Beginning at room temperature (prior to $t = 0$), the temperature ramps at ~2 °C s$^{-1}$ to the experimental temperature ($T_{exp}$; black dashed line). The start of the experiment ($t = 0$) begins once the temperature oscillations about $T_{exp}$ are < 2 °C. The profile in **(a)** is an example of the minimum temperature used (51.2 °C), and in **(b)** is an example of the maximum (104.3 °C).

At the start of the viscosity measurements, the amorphous solid particles were non-spherical (e.g. Fig. 4a, 4c and Fig. 5a, 5c). If the temperature used in the experiment ($T_{exp}$) was hot enough, the material began to flow to minimize the



surface energy of the system during the experiment. To relate the flow of the particle to the particle's viscosity, a quantitative measure of the shape change was needed.

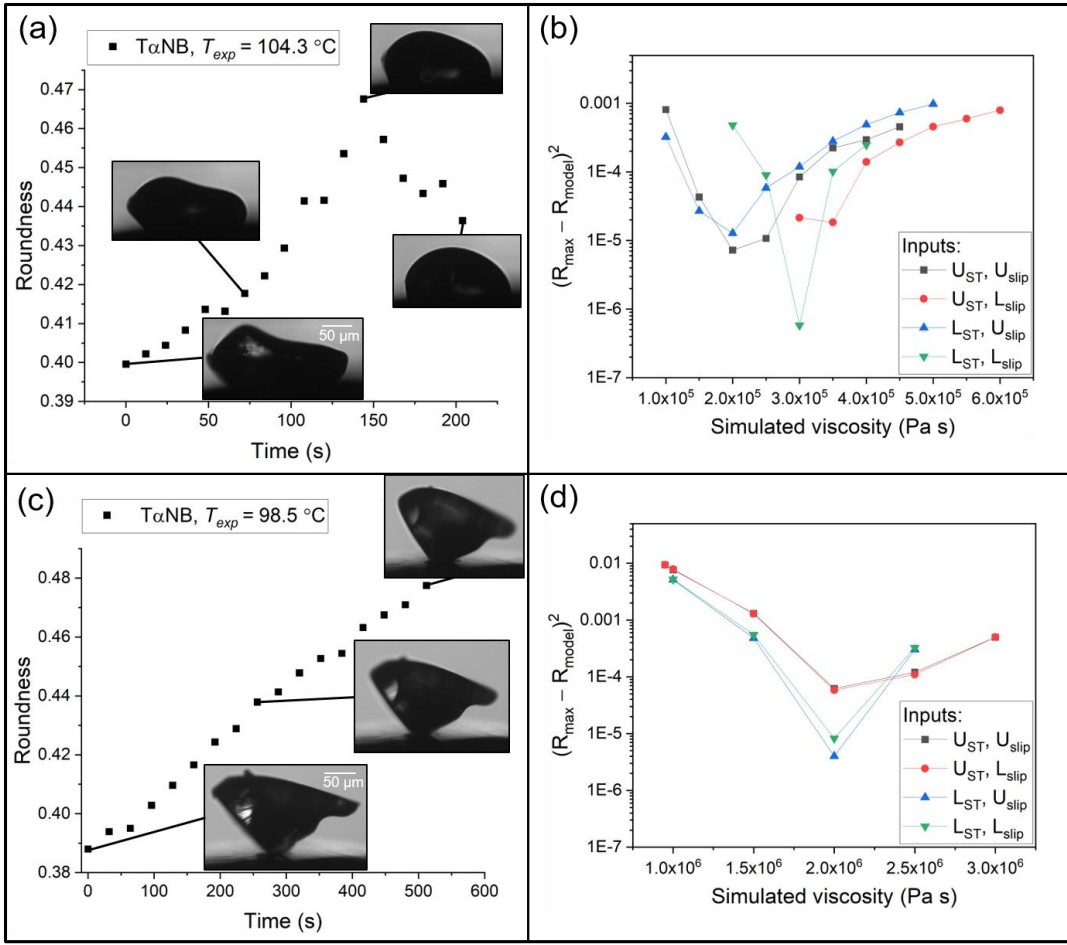

**Figure 4** Hot-stage microscopy experiments with TαNB. Roundness as a function of time at **(a)** 104.3 °C and **(c)** 98.5 °C. The squared difference between observed and simulated roundness as a function of simulated viscosity for the two sample TαNB experiments, where **(b)** corresponds to **(a)**, and **(d)** to **(c)**. The simulation that best reproduces the observation corresponds to the minimum sum of squared difference. The different colors correspond to different input combinations of surface tension and slip length used in the simulations, where $U_{ST}$, $U_{slip}$, $L_{ST}$, and $L_{slip}$ correspond to the upper limit of surface tension, the upper limit of slip length, the lower limit of surface tension, and the lower limit of slip length, respectively.

To quantify the shape change we used roundness, which is a measure of how close the shape is to a circle (a roundness value of 1 indicates a perfect circle). Roundness was calculated with the following equation:

$$\text{Roundness} = \frac{4A}{\pi\left(L_{Feret,max}\right)^2} \, ,$$

(1)





where A is the projected area of the particle and $L_{Feret,max}$ is the maximum Feret length of the 2-D projection, which is defined as the maximum distance between two parallel tangential planes to the projected 2-D geometry. The image analysis software ImageJ was used to process the images recorded during the experiments. The silhouette of the needle was removed from the images, and then the images were subsequently binarized. The area and $L_{Feret,max}$ were determined from the sequence of binary images using a MATLAB script.


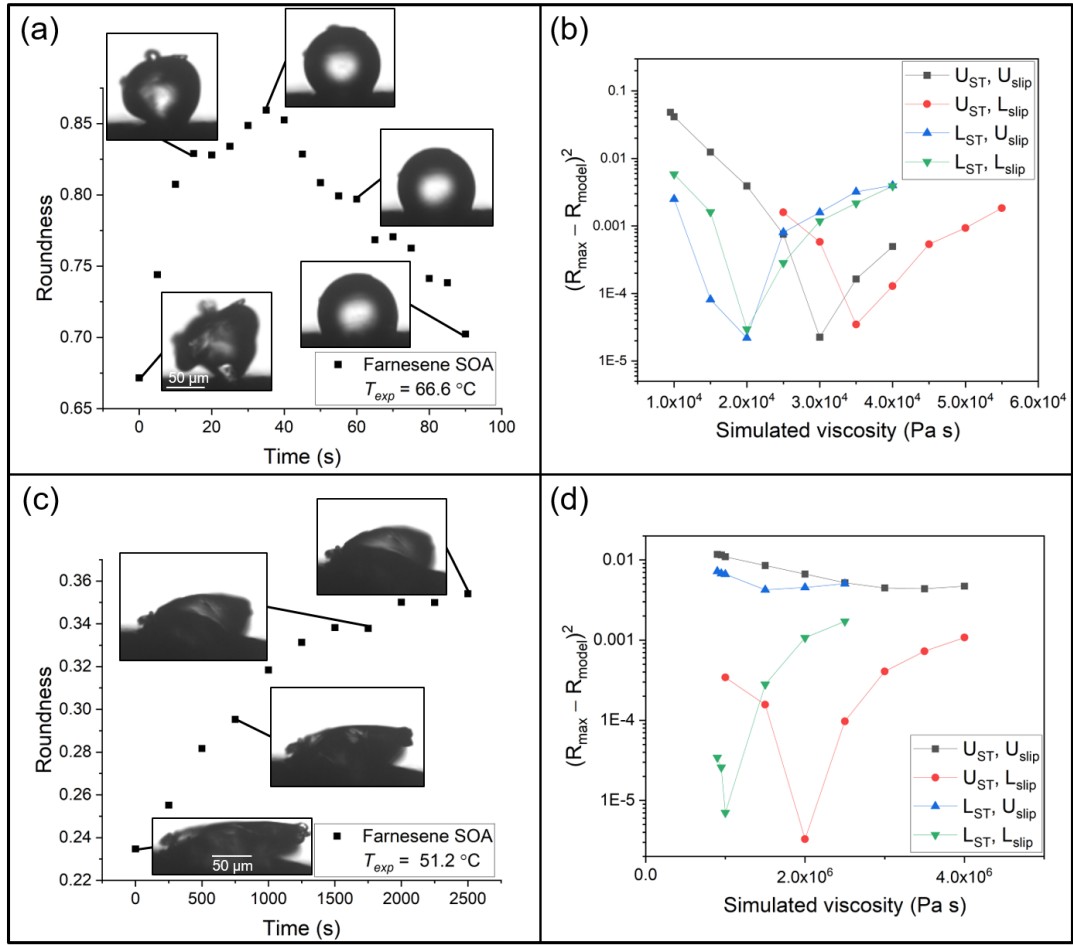

**Figure 5** Hot-stage microscopy experiments with farnesene SOA. Roundness as a function of time at **(a)** 66.6 °C and **(c)** 51.2 °C. The squared difference between observed and simulated roundness as a function of simulated viscosity for the two sample farnesene SOA experiments, where **(b)** corresponds to **(a)**, and **(d)** to **(c)**. The simulation that best reproduces the observation corresponds to the minimum sum of squared difference. The different colors correspond to different input combinations of surface tension and slip length used in the simulations, where $U_{ST}$, $U_{slip}$, $L_{ST}$, and $L_{slip}$ correspond to the upper limit of surface tension, the upper limit of slip length, the lower limit of surface tension, and the lower limit of slip length, respectively.






In HSM experiments with TαNB and farnesene SOA, if the temperature was hot enough for the material to flow, the roundness of the particles first increased to minimize the surface area of the particle and then decreased as the material spread on the surface (e.g. Fig. 4a and Fig. 5a). From the images, we determined the maximum roundness ($R_{max}$) and the time necessary to reach $R_{max}$, defined here as the experimental flow time, $\tau_{exp,flow}$. The $\tau_{exp,flow}$ and $R_{max}$ values were then used in the fluid dynamics simulations discussed below to determine the viscosity. In some cases, with TαNB, crystallization occurred before the maximum roundness was observed. In these experiments, $\tau_{exp,flow}$ was assigned as the time just before crystallization was observed and the roundness at $\tau_{exp,flow}$ was assigned $R_{max}$.

**2.4 Fluid dynamics simulations**

To extract viscosity values from the experiments, the physics software COMSOL Multiphysics (v5.4) was used. The Microphysics Module within COMSOL was used along with a two-phase laminar flow and moving mesh. Only the 2-D silhouette of the particle could be imaged; the 3-D morphology of the particles was not known. Therefore, a 2-D model was used to represent particle shape in the simulations which has the added benefit of reduced simulation times. Sensitivity tests show that viscosities determined with a 2-D model are similar to viscosities determined with a 3-D model (See Section 3.3). The initial 2-D geometry used in the simulations was the same as the projected geometry observed in the HSM experiments. To obtain the same initial 2-D geometry, the initial projected geometry captured during the HSM experiment was binarized and imported into COMSOL. For the simulations, slip length, surface tension, and contact angles were needed. For slip lengths and surface tensions, conservative upper and lower limits were estimated based on literature values (Table 1). For contact angles, 37.5° and 29.9° were used for TαNB and farnesene SOA, respectively, based on measurements (see Section 2.5).

**Table 1.** Input values of surface tension and slip length used in fluid dynamics simulations.

| Material | Surface tension (N m$^{-1}$) | Navier slip length (nm) |
|----------|------------------------------|-------------------------|
| TαNB | 0.0486–0.0546[a] | 5–1000[b] |
| Farnesene SOA | 0.023[c]–0.045[d] | 5–1000[b] |

[a] The surface tension of TαNB was reported to be 0.0516 N/m, based on the ACD/Labs Percepta Platform-PhysChem Module retrieved from ChemSpider. The upper and lower limits correspond to the reported uncertainty of ± 0.0030 N/m.

[b] The range of values for Navier slip length is based on values reported in the literature (Baudry et al., 2001; Bhushan et al., 2009; Cho et al., 2004; Churaev et al., 1984; Cottin-Bizonne et al., 2002, 2005; Craig et al., 2001; Jing and Bhushan, 2013; Joseph and Tabeling, 2005; Mcbride and Law, 2009; Tretheway and Meinhart, 2002; Vinogradova et al., 2009; Zhu et al., 2012).

[c] The lower limit for the surface tension of farnesene SOA was the lower uncertainty of the ACD/Labs value for farnesene retrieved from ChemSpider.

[d] The upper limit for the surface tension of farnesene SOA is an upper limit to surface tensions reported in the literature for water + SOA solutions and common SOA functional groups such as alcohols, organic acids, esters, and ketones (Demond and Lindner, 1993; Gorkowski et al., 2020; Gray Bé et al., 2017; Hritz et al., 2016).

The time used in the simulations, $\tau_{model}$, was set to $\tau_{exp,flow}$. The output of a simulation was a binary 2-D image of the final geometry at $\tau_{model}$. The roundness of the final 2-D geometry, $R_{model}$, was then calculated with the same MATLAB script as used for the images recorded during HSM experiments (see above). $R_{model}$ was then compared to $R_{max}$ by calculating the squared difference between $R_{model}$ and $R_{max}$. The viscosity in the simulation was then varied and the squared difference between $R_{model}$ and $R_{max}$ was determined as a function of the viscosity. This procedure was repeated using upper and lower limits for the surface tension and upper and lower limits for the slip length, resulting in four plots of the squared difference between $R_{model}$ and $R_{max}$ as a function of viscosity for each experiment (e.g. Fig. 4b, 4d and Fig. 5b, 5d). From each plot, the viscosity that gave the smallest squared difference between $R_{model}$ and $R_{max}$ was identified, resulting in four viscosities for each experiment. The largest and smallest viscosity from these four minima was assigned as the upper and lower limits for the experiment. This whole process was then repeated for several experiments. The upper limit was assigned as the average of the upper limit values from replicate experiments summed with twice the standard error of the mean. The lower limit was assigned as the average of the lower limit values subtracted by twice the standard error of the mean.

### 2.5 Contact angle measurements

For the fluid dynamics simulations discussed above, the contact angle between the material studied and the substrate was needed. To determine these contact angles, the material was heated to the point where it had completely liquefied, flowed, and reached an equilibrium geometry on the flat end of the ultra-fine needle (e.g. Fig. S3). The equilibrium contact angle between the material and the flat end of an ultra-fine tungsten needle was then measured directly from captured images using the ImageJ angle tool. Based on these measurements, the contact angles for TαNB and farnesene SOA were $37.5 \pm 3.6°$ and $29.9 \pm 3.8°$, respectively, where the uncertainties are twice the standard error of the mean. The simulations described above used the average contact angles determined with this approach. However, changing the contact angle in the simulations by $\pm 2\sigma$ only changed the viscosity in the simulations by 5% at most, which was considered small compared to the uncertainties in the reported viscosities.

### 2.6 Measurements of the chemical composition of farnesene SOA using mass spectrometry

To determine the chemical composition of the SOA, the SOA was generated in the environmental chamber using the same procedure as discussed above and then collected onto aluminum foil, followed by extraction in 1:1 (v/v) LCMS grade acetonitrile and water. The chemical composition of the extracted SOA was obtained in positive and negative ionization mode using ultra performance liquid chromatography-photodiode array detection-electrospray ionization-high resolution mass spectrometry (UPLC-PDA-ESI-HRMS). A Vanquish Horizon UPLC (Thermo Scientific) was equipped with a Luna 1.6 μm Omega Polar C18 (150 x 2.1 mm column). Solvents A and B used during liquid chromatography were HPLC grade acetonitrile (containing 0.1 % v/v formic acid) and water (also with 0.1 % v/v formic acid), respectively. A Q-Exactive Plus Orbitrap mass spectrometer (Thermo Scientific) with a mass resolving power of ~1.4 x $10^5$ at 200 m/z was used and data were acquired from 50-750 m/z. The ESI source parameters were set to a spray voltage of +3.5 kV in positive mode and -2.5 kV in negative mode, the capillary temperature was 320 °C, and the S-lens ion funnel RF level was 50. In addition to the samples, a solvent blank was prepared following the





same procedure above, but using a clean aluminum foil substrate without analyte. The analysis procedure of the mass spectrometry data has been described in detail previously (Maclean et al., 2021a; Smith et al., 2021). Briefly, the peaks were integrated and extracted from the mass spectra from 2–16 minutes corresponding to SOA elution in the chromatogram, and peaks containing 13C isotopes were removed. All peaks were assigned to the formulas $C_xH_yO_z$
with an accuracy of 0.0005 m/z units while containing the assignments to closed-shell ions with even nominal masses and constraining H/C to 0.30–2.25 and O/C to 0.00–2.30. Masses were only considered if they were three times more abundant in the sample compared to the blank. The assigned ion formulas were corrected for the ionization mechanism, and all the HRMS results below are reported as formulas of neutral SOA compounds. The assumed ionization mechanisms were the formation of adducts with $H^+$ or $Na^+$ for positive ions and deprotonation for negative ions.

**2.7 Predictions of the temperature-dependent viscosity of farnesene SOA using the chemical composition**

The temperature-dependent viscosity of farnesene SOA was predicted from the chemical composition of farnesene SOA using the method in DeRieux et al. (2018). First, the $T_g$ of each compound identified in the HRMS was determined using Eq. 2, which relates the number of C, H, and O in a compound, $i$, to its $T_g$ (DeRieux et al., 2018):

$$T_{g,i} = \left(n_C^0 - \ln(n_C)\right)b_C + \ln(n_H)b_H + \ln(n_C)\ln(n_H)b_{CH} \\ + \ln(n_O)b_O + \ln(n_C)\ln(n_O)b_{CO} \tag{2}$$

Where $n_C$, $n_H$, and $n_O$ are the number of carbon, hydrogen, and oxygen, respectively. The coefficients $n_C^0$, $b_C$, $b_H$, $b_{CH}$, $b_O$, and $b_{CO}$ were 12.13, 10.95, −41.82, 21.61, 118.96, and −24.38, respectively, for CHO-containing compounds.

Next, the $T_g$ of the overall SOA (for dry conditions) was calculated using the Gordan-Taylor equation (Gordan and Taylor, 1952) with an assumed Gordan-Taylor constant ($k_{GT}$) of 1:

$$T_{g,\,org} = \sum_i w_i T_{g,i} \tag{3}$$

Where $w_i$ is the mass fraction of component $i$. The $w_i$ values were assumed to be proportional to the relative abundance ($I_i$) in the combined mass spectra, that is:

$$w_i = I_i \tag{4}$$

As described previously, this assumption is a limitation of these viscosity predictions (DeRieux et al., 2018; Song et al., 2019). The Vogel-Fulcher-Tammann equation was used to describe the temperature dependence of viscosity
(Fulcher, 1925):

$$\log_{10} \eta = A + 0.434 \frac{D_f T_0}{T - T_0} \tag{5}$$

where $\eta$ is viscosity, $T$ is temperature, $D_f$ is the fragility parameter, $T_0$ is the Vogel temperature, and $A = -5$ (Angell, 1991). We assumed $D_f = 10$, as done previously (Gervasi et al., 2020; Maclean et al., 2021a, 2021b; Shiraiwa et al., 2017b). The Vogel temperature, $T_0$, was determined from Eq. 6 below, which is derived from Eq. 5, assuming the
330 viscosity is $10^{12}$ Pa s at $T = T_g$ (Angell, 1991, 2002):



$$T_0 = \frac{39.17\,T_{\mathrm{g}}}{D_f + 39.17} \tag{6}$$

where $T_{\mathrm{g}}$ was obtained from Eq. 3 and $D_f$ was set to 10. Once $T_0$ was calculated, a VFT curve was constructed using Eq. 5.

## 3 Results and discussion

### 3.1 Measured viscosities of TαNB

The viscosity of TαNB was determined at two temperatures: 98.5 to 104.3 °C. At each temperature, the experimental flow time, $\tau_{\mathrm{exp,flow}}$, and maximum roundness, $R_{\mathrm{max}}$, was determined for three different TαNB particles (Table S1). On average, the $\tau_{\mathrm{exp,flow}}$ value decreased as temperature increased, as expected. The 2-D fluid dynamics simulations (Section 2.4) were used to convert $\tau_{\mathrm{exp,flow}}$ and $R_{\mathrm{max}}$ values to viscosities. Upper and lower limits for the viscosities were determined from each experiment and combined to give an overall upper and lower limit to the viscosity of the material (Fig. 6). The viscosity midpoints at 98.5 and 104.3 °C were $1.1 \times 10^6$ and $2.8 \times 10^5$ Pa s, respectively. For reference, the room temperature viscosities of lard and tar pitch are $10^3$ and $10^8$ Pa s, respectively. The viscosity midpoints of TαNB decreased by a factor of ~4 for a 5.8 °C increase in temperature.

The measured viscosities were compared to literature values from Plazek et al. (1999) by applying a VFT fit to their data, indicated by the blue line in Fig. 6 and the light blue shading representing the 95% confidence intervals of the fit. The viscosity measured using our HSM method agrees with the viscosity from Plazek et al. (1999), within the uncertainty of our measurements and the VFT fit to their data, suggesting our approach is valid for determining temperature-dependent viscosities of small samples (< 1 mg of material).

The temperature range over which we reported viscosities for TαNB was quite narrow (98.5 to 104.3 °C). The upper limit of the temperature range in our experiments is limited by flow becoming so fast that $R_{\mathrm{max}}$ is reached before the temperature has stabilized (oscillations < 2 °C) at $T_{\mathrm{exp}}$. At $T_{\mathrm{exp}} = 104.3$ °C, $R_{\mathrm{max}}$ is reached in ~1 minute, so the upper-temperature range in our experiments for TαNB is likely only a few degrees warmer. The lower limit of the temperature range for TαNB experiments was restricted, in part, because crystallization became dominant at colder temperatures. Experiments at ~97 °C were attempted, but crystallization occurred before significant flow was detected. However, a wider temperature range should be possible for other amorphous materials where crystallization does not occur.

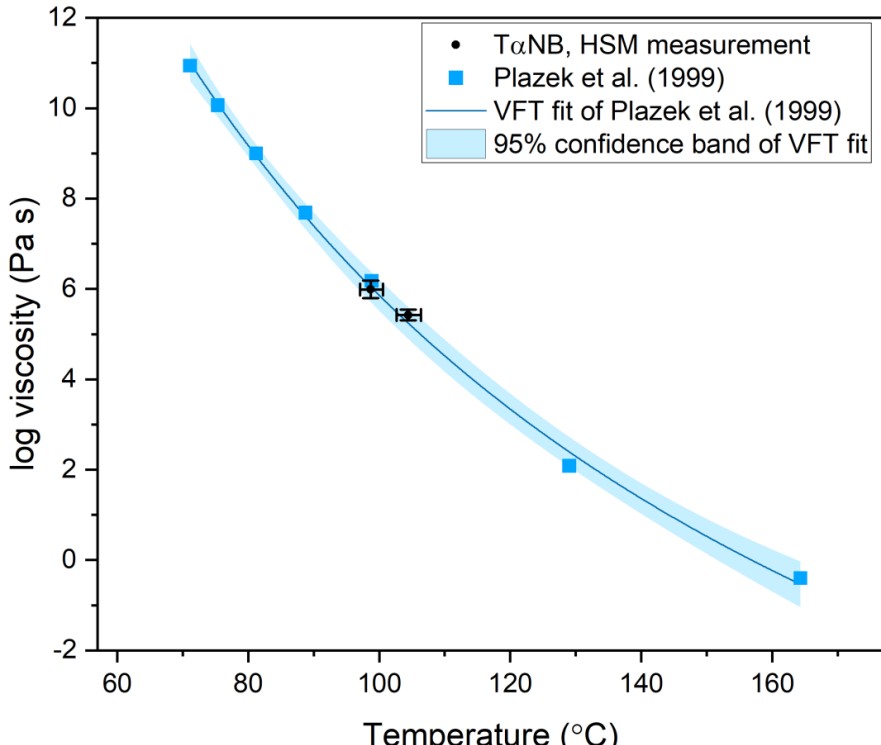

**Figure 6.** The viscosity of TαNB as a function of temperature. The blue squares are measurements from Plazek et al. (1999). The blue trend line is a Vogel-Fulcher-Tammann (VFT) fit to their data, with the light blue area corresponding to the 95% confidence band of the fit. The black data points are the midpoint of the upper and lower limits measured with the hot-stage microscopy (HSM) technique. The y-error bars correspond to the upper and lower limits of viscosity from three separate experiments (95% confidence interval). The temperature uncertainty (x-error) is a propagation of the uncertainty in the temperature control stage calibration and the largest temperature oscillation during the experiment.

### 3.2 Measured viscosity of farnesene SOA

Viscosities for farnesene SOA were determined at temperatures of 51.2, 61.5, and 66.6 °C. For each experiment, $\tau_{exp,flow}$, $R_{max}$, and the upper and lower limits to the viscosities were determined using the observed projected geometry and 2-D fluid dynamics simulations (Table S2). On average, $\tau_{exp,flow}$ increased with colder temperature: as the temperature decreased by ~15 °C, $\tau_{exp,flow}$ increased by ~2 orders of magnitude. At $T_{exp} = 66.6$ °C, the temperature was near the upper limit of the measurable temperature range, since $\tau_{exp,flow}$ was ~30 seconds at this temperature. It could be possible to do experiments a couple of degrees warmer, but soon the flow will be so fast that $R_{max}$ is reached before the temperature stabilized (oscillations < 2 °C) at $T_{exp}$. On the lower end, we chose $T_{exp} = 51.2$ °C for the practical purpose of keeping $\tau_{exp,flow}$ less than a few hours. In principle, the $\tau_{exp,flow}$ in our experiments could be as long as multiple days.





The viscosity midpoints determined from the fluid dynamics simulations for farnesene SOA at $T_{exp}$ = 51.2, 61.5, and

66.6 °C were $3.4 \times 10^6$, $1.0 \times 10^5$, and $2.6 \times 10^4$ Pa s, respectively (Fig. 7).  These viscosities fall between the viscosities

of tar pitch (viscosity = $10^8$) and lard (viscosity = $10^3$).

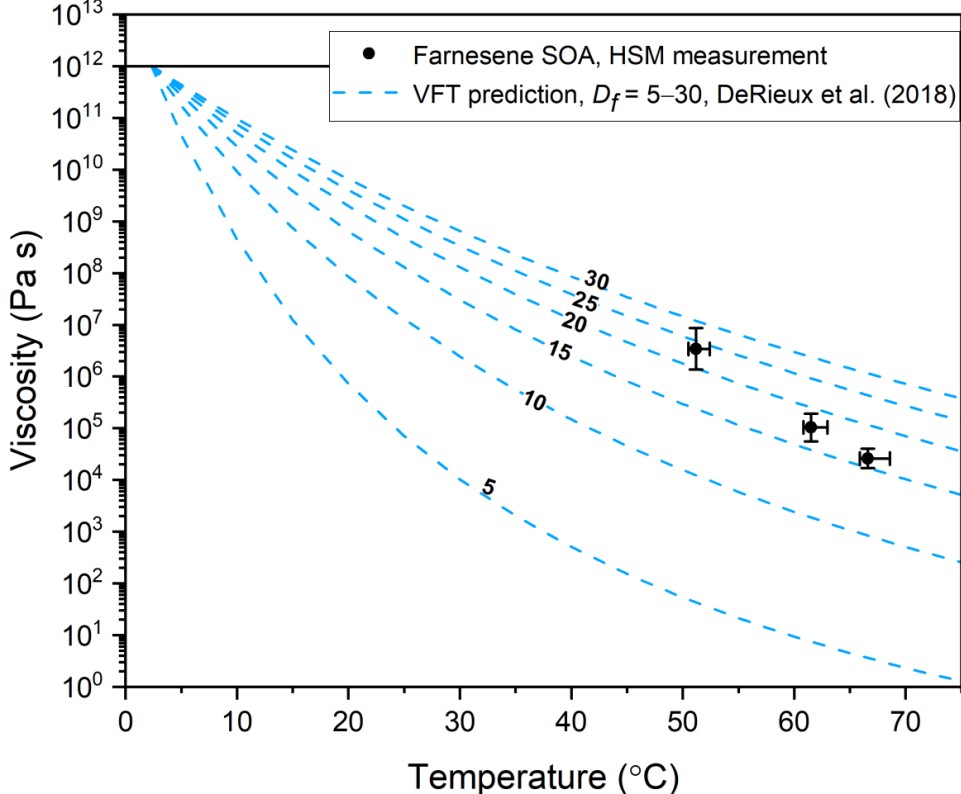

**Figure 7.**  Viscosity as a function of temperature for farnesene SOA. The black data points represent the midpoint of the upper
and lower limits of viscosity, measured with hot-stage microscopy (HSM). The $y$-error bars correspond to the upper and lower
limits of viscosity from at least two separate experiments (95% confidence interval).  The temperature uncertainty ($x$-error) is a
propagation of the uncertainty in the temperature control stage calibration and the largest temperature oscillation during the
experiment. Note that the $x$-error bars are asymmetric because the largest temperature oscillation was always toward hotter
temperatures. The blue dashed lines are predictions using the method in DeRieux et al. (2018) with HRMS data, and $D_f$ values
ranging from 5–30 (labelled on each dashed line). The solid black line represents the viscosity of a glass, $10^{12}$ Pa s.

### 3.3 Sensitivity to the geometry assumed in the fluid dynamics simulations

All the viscosity results discussed above were determined using fluid dynamic simulations and a 2-D model to describe

the HSM experiments. A 2-D model was used to minimize computation time and because the actual 3-D morphology

of the particles was not known. To test if the simulated viscosities were sensitive to the assumed morphology (2-D vs

3-D), we compared simulations with both 2-D and 3-D geometries for a few experimental conditions. To create a 3-

D geometry, the 2-D geometry was expanded into 3-D by using the "extrude" function in COMSOL (e.g. Fig. S4). To





determine the extrude depth, we used the conservation of mass. Specifically, we assumed the depth of the extrusion was equal to the particle volume (V) divided by the initial 2-D area ($A_i$). $A_i$ was measured directly using ImageJ. V

was determined from images of the particles after they had flowed and reached an equilibrium geometry, assuming a spherical cap geometry. The difference in simulated viscosity between the 2-D and 3-D simulations was minor (Table S3). Specifically, for TαNB, the upper and lower limits of viscosity calculated using the 3-D geometry fell within the upper and lower limits of the 2-D geometry. For farnesene SOA, the upper and lower limits of viscosity were equivalent for 2-D and 3-D. Based on the 2-D vs 3-D results for both TαNB and farnesene SOA, 2-D simulations were

deemed adequate, since there were only minor variations in simulated viscosity and there was a significant amount of saved computation time (~5 seconds for each 2D simulation compared to ~7 – 45 minutes for each 3-D simulation).

### 3.4 Predicted viscosities of farnesene SOA based on chemical composition

The combined mass spectra for farnesene photooxidation SOA in positive and negative mode are shown as a function of neutral mass and normalized to the largest peak in each mode (Fig. S5). The chemical composition of the SOA is

discussed in Section S6 of the supplement, including a summary of compounds that have been previously identified in the literature (Table S4). Based on the HRMS data and Eq. 3, the glass transition temperature for farnesene SOA was predicted to be $T_g = 275$ K. This value, together with Eq. 5 and 6, was used to predict the viscosity of farnesene SOA (blue curves in Fig. 7). The predicted viscosities were ~1–2 orders of magnitude less than the measured viscosities. One possible reason for this discrepancy could be the assumed $D_f$ in the calculations; we assumed a $D_f$

value of 10 in the original predictions as done previously (DeRieux et al., 2018; Gervasi et al., 2020; Maclean et al., 2021a, 2021b; Shiraiwa et al., 2017b). As shown by Angell (1997), and discussed in Gervasi et al. (2020), the $D_f$ value for organics is typically in the range of ~5–30 (Angell, 1997; Gervasi et al., 2020). In Fig. 7, we show viscosity predictions for this range of $D_f$ values. Predictions using $D_f$ of approximately 15 to 25 are in better agreement with our results, although no single $D_f$ value can explain all our data. Equation 4 assumes that the mass fraction of component

$i$ in the SOA is proportional to the relative abundance ($I_i$) in the combined mass spectra, which is known to be a limitation in viscosity predictions (DeRieux et al., 2018; Maclean et al., 2021b; Song et al., 2019). Previous studies have also used an adjusted mass approach based on the work of Nguyen et al. (2013) to predict the relationship between the mass fraction of component $i$ and the relative abundance in the combined mass spectra (Maclean et al., 2021b; Nguyen et al., 2013). We have also tried this approach, but obtained worse agreement between the measured and

predicted viscosities.

Another possible explanation for discrepancies seen between the experimental and viscosity predictions is possible in-source fragmentation occurring for farnesene SOA during the UPLC-PDA-ESI-HRMS analysis. In-source fragmentation could lead to smaller molecular weight compounds seen in the mass spectra < 250 Da and the presence of compounds containing less than 10 carbons (Fig. S6). These lower molecular weight compounds would lead to

lower calculated glass transition temperatures and therefore contribute to an overall lower predicted viscosity than the experimentally determined viscosity values. Future studies are needed in order to confirm if this phenomenon is happening for farnesene SOA.



## 4 Conclusions

We have presented the details of a new hot-stage microscopy technique for measuring the temperature-dependent viscosity of samples with small sizes (< 1 mg of material). Highly viscous samples were stuck to a flat-tipped needle, heated in a temperature control stage, and imaged with a microscope. When the temperature was increased, the particles flowed to minimize their surface energy. The viscosity was extracted from these images by modeling the flow using fluid dynamics simulations. The technique was validated by reproducing the viscosity of a literature standard, TαNB, at 98.5 and 104.3 °C. As an application to the study of atmospheric aerosols, the viscosity of farnesene SOA was measured at 51.2, 61.5, and 66.6 °C. The viscosity of farnesene SOA increased by ~2 orders of magnitude as the temperature decreased by ~15 °C. The DeRieux et al. (2018) parameterization has been previously used to predict the viscosity of SOA from its chemical composition. We compared our measurements here to the predicted viscosity using HRMS measurements with the DeRieux et al. (2018) parameterization. Using the previously used value of $D_f = 10$, the parameterization under predicted our measurements by ~1–2 orders of magnitude. Using higher $D_f$ values improved agreement between measurement and prediction, but no single value of $D_f$ could reproduce our measurements. Other possible explanations for the discrepancy between prediction and measurement was in-source fragmentation of farnesene SOA during HRMS measurement. Nevertheless, this study illustrates that our new experimental approach for measuring viscosities as a function of temperature provides important data for testing methods used for predicting viscosities of SOA in the atmosphere.

*Data availability.* Underlying material and related items for this paper are located in the Supplement.

*Author Contributions.* KJK and AKB came up with the idea, methods, and experimental design. KJK performed the calibrations, viscosity measurements, and analysis. KLM designed and wrote the image analysis script. KJK and SX designed the fluid dynamics simulations. NRS and SAN performed the HRMS experiments and analysis. KJK and AKB led the writing of the paper, with contributions from NRS and SAN. All authors gave their final approval for publication.

*Competing interests.* The authors declare that they have no conflict of interest.

*Financial support.* This work was supported by Natural Sciences and Engineering Research Council of Canada (NSERC) through grant RGPIN/04441-2016, and the US National Science Foundation through grant AGS-1853639. The high-resolution mass spectrometer instrument used in this work was purchased with grant NSF CHE-1920242.



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
