# Peer review of "A New Hot-Stage Microscopy Technique for Measuring Temperature-Dependent Viscosities of Aerosol Particles and its Application to Farnesene Secondary Organic Aerosol"

_Atmospheric Measurement Techniques, 2022_

## Author Response (AR1)

Dear Prof. Francis Pope,

Given below are our responses to the comments provided by the reviewers of our manuscript. For clarity, the referee comments are in black text, and are preceded by bracketed, italicized numbers (e.g. *[1]*). Our (authors') responses are in in blue text below each comment or question with matching italicized numbers (e.g. *[A1]*). The revised text according to the referees' comment or question is in green text below each authors' response and line numbers [e.g. L167] refer to the revised manuscript (version without tracked changes). We thank the referees for their time and care reading our manuscript and for their very helpful comments.

Sincerely,

Allan Bertram
Professor of Chemistry
The University of British Columbia

Reviewer: 1

This work describes a new method for measuring the viscosity of aerosol particles as a function of temperature. The authors combine a hot-stage microscopy method with fluid dynamics simulations and validate their method with a known standard. When applying the method to lab-generated SOA, viscosity values that were 1-2 order of magnitude higher than predictions were observed. This highlights the need for viscosity measurements of complex systems, and motivates the publication of this research.

*[1]* While the method appears to work as applied to one validation system, it is hard to see the generality of this method as a tool for viscosity characterization without further validation. The authors report just 2 data points and conclude the method is valid, and go on to show orders of magnitude differences in a sample system (compared to estimates based on models). In my opinion, more validation systems need to be explored before the applicability of this method can be established.

*[A1]* Thank you for the feedback. To address the referee's concerns we have included measurements at three temperatures of a second validation system, phenolphthalein dimethyl ether. The details of these measurements have been added to all relevant sections and figures, and a new figure (Fig. 5) shows example experiments and simulation data. At the temperatures measured, our hot-stage measurements agree with the literature viscosity of PDE within the respective uncertainties of the data.

In addition to my general critique above, I have some specific comments:

*[2]* Can this method work for substances at room temperature or below? What is the viscosity range that is accessible to this technique?

*[A2]* Good questions. The current method cannot be used to provide measurements of viscosity at room temperature. In theory, the current method could be used to provide measurements of viscosity at 290−300 K if the sample was prepared in a cold-room, but we have not explored this option. The viscosity range that is accessible to this technique is roughly $10^4$ to $10^8$ Pa s. Higher viscosity may be attainable if experiments were carried out over multiple days. To address the referee's comment, the following sentences was added to the abstract and conclusions [L15, L478]:

"The current method can be used to determine viscosities in the range of roughly $10^4$ to $10^8$ Pa s at temperatures greater than room temperature. Higher viscosities may be measured if experiments are carried out over multiple days."

*[3]* It seems that the technique could equally measure the melting temperature - can this be related to Tg and then correlated with a viscosity? This would bypass the need for fluid dynamics and may provide a simpler method for determining Tg than the traditional DSC methods. Is there an advantage to obtaining viscosity, which is then used to determine Tg anyway, in order to derive viscosity across the range of temperatures?

*[A3]* When the referee is referring to the melting temperature, are they referring to the crystalline solid to liquid transition? We can certainly measure this transition for crystalline material using the current apparatus, but this would not be useful for secondary organic aerosol. Secondary organic aerosol consists of thousands of compounds, and the concentration of any given compound is not expected to be high enough for crystallization. To address the referee's comment, we have added the following to the revised manuscript [L165]:

"This method of collection resulted in droplets with spherical cap geometries and diameters of ~30 to 250 µm, formed by the aggregation of smaller SOA particles on the hydrophobic glass substrates. These droplets are not expected to contain crystalline material, since SOA consists of hundreds of different compounds, and the concentration of any given compound is not expected to be high enough for crystallization (Marcolli et al., 2004). Furthermore, optical images of SOA particles are consistent with a non-crystalline state for the SOA."

*[4]* The authors refer to this as a method for measuring aerosol particle viscosity, whereas in fact it is measuring the viscosity of aerosol material (they are no longer individually resolved particles).

*[A4]* Thank you for the comment. Where applicable, we have changed the wording from "farnesene SOA" to "farnesene SOA material".

*[5]* It is unclear to me if the sample is on a needle or a slide? From the schematic, the images are looking from below, so is the particle on the side of the needle? What is the effect of gravity on the shape of this system?

*[A5]* Figure 2 has been slightly modified to better illustrate that the particle is attached to the needle which overhangs (but does not come into contact with) the viewing window.

The effect of gravity on the shape of the particle is expected to be negligible because the force due to gravity ($F_g$) acting on the particle is much smaller than the surface tension force ($F_{ST}$) acting on the particle. To address the referee's comment, we have added a calculation to the Supplemental to illustrate the difference in these two forces (see Supplemental, Section S3).

We have also added the following to the main text [L243]:

"Note, in these experiments, the effect of gravity on the shape of the particles is expected to be negligible because the force due to gravity ($F_g$) acting on the particle is much smaller than the surface tension force ($F_{ST}$) acting on the particle (See Supplemental, Section S3)."

The following is the information was added to the Supplemental, Section S3:

"To illustrate that the effect of gravity on the shape of the particles is expected to be small, we compared the force of surface tension ($F_{ST}$) and the force of gravity ($F_g$) acting on a particle. The force of surface tension can be calculated with the following equation:

$$F_{ST} = 2\sigma\pi r$$

Where $\sigma$ is the surface tension and $r$ is the radius of the droplet.

The force of gravity can be calculated with the following equation:

$$F_g = mg = \left(\frac{4}{3}\pi r^3 \rho\right) g$$

Where $r$ is the radius of the droplet, $\rho$ is the density, and $g = 9.8$ m$^2$ s$^{-1}$.

Assuming a particle diameter of 190 μm (the largest particle diameter from our experiments), a surface tension of 0.023 N m$^{-1}$ (the smallest surface tension in our experiments), and a density of 1200 kg m$^{-3}$, $F_{ST}$ is approximately a than a factor of 325 greater than $F_g$. The ratio of $F_{ST}$ to $F_g$ will be even larger for smaller droplets and larger surface tensions."

Reviewer: 2

Kiland et al. show how a microscope interfaced with a hotplate can be used to measure the temperature dependence of viscosity for samples where less than 1 mg of material is available. The technique is compared against one substance with known temperature dependence of viscosity. It is further shown that the technique can be applied to secondary organic material collected from an environmental chamber experiment. These data are discussed in the context of parameterization-to-data comparison.

The presented technique is a new addition to the toolbox of scientists who are interested in measuring the viscosity of organic aerosols. The underlying techniques are sound. I am excited to see this work being pursued and I recommend the paper for publication. However I have some suggestions on how the data interpretation might be improved.

Comments:

*[1]* The technique focuses on bulk samples. Therefore secondary organic material should be used throughout to distinguish it from aerosol sampling.

*[A1]* Thank you for the comment. Where applicable, we have changed the wording from "farnesene SOA" to "farnesene SOA material".

*[2]* TαNB was heated above the melting point of ~185°C and then cooled to generate glassy material. This step is not mentioned for the SOM. Presumably, the particles turned glassy in the chamber or the -20°C freezer. Was heating of the SOM avoided due to the possibility of evaporation of more volatile components, thermal decomposition, and thermally induced chemical reactions? Furthermore, not all crystalline substances can be heated above the melting point without decomposition. Presumably this is the reason why sucrose - a seemingly more logical choice to validate the system due to its widespread use in the community - was not used. It would be helpful to add some discussion that highlights the assumptions and limitations associated with the sample preparation requirements.

*[A2]* The standards used in our work, TαNB and phenolphthalein dimethyl ether (PDE, added in the revised manuscript) were both in a crystalline state when purchased. A heating and cooling step was used to convert these chemicals to an amorphous highly viscous state at room temperature. A similar step was not needed for the SOA material, since the SOA material was already in an amorphous highly viscous state at room temperature when generated.

Yes, one of the reasons for not using sucrose as a standard was due to the possibility of thermal decomposition of the material when convert the crystalline material to an amorphous state using a heating and cooling cycle.

In the revised manuscript, we have significantly expanded the discussion on the assumptions and limitations for sample preparation [L133]:

"To be compatible with our new technique, the material must have the following properties: 1) the material must be amorphous (i.e. non-crystalline), 2) the material must have a viscosity greater than approximately $10^8$ Pa s at room temperature so the material does not flow significantly on a timescale of hours at room temperature, 3) the material must be in the form of particles with dimensions ~20 – 300 µm, and 4) the particles must be in a non-equilibrium geometry at the start of the experiments. Below is a description of how we generated particles with these properties for TαNB, PDE, and farnesene SOA material.

When purchased, 1,3,5-tris (1-naphthyl)benzene (TRC Canada) and phenolphthalein dimethyl ether (Polymer Source) were in a crystalline state. To generate amorphous solid particles of TαNB and PDE with the properties discussed above, the following procedure was used: First some of crystalline material was placed on a glass slide, and the temperature of the slide was increased to a temperature above the melting point of the crystalline material (TαNB: mp = 184 – 186 °C; PDE: mp = 100 °C). After the crystalline material had melted, the temperature was rapidly decreased (~2 °C s$^{-1}$) to room temperature, resulting in the formation of a glassy material (Magill and Plazek, 1967; Plazek and Magill, 1966; Stickel et al., 1996). The glassy material was then scraped off the slide with a razor blade, forming amorphous solid particles in a non-equilibrium geometry (i.e. jagged geometry) with area-equivalent diameters ranging from ~60 to 190 µm. The formation of a glass state was verified by showing that the material flowed at a temperature significantly lower than the melting temperature of the crystalline material (see below)."

And [L165]:

"(Cytonix USA). Samples were collected for 3 hours. This method of collection resulted in droplets with spherical cap geometries and diameters of ~30 to 250 µm, formed by the aggregation of smaller SOA particles on the hydrophobic glass substrates. These droplets are not expected to contain crystalline material, since SOA consists of hundreds of different compounds, and the concentration of any given compound is not expected to be high enough for crystallization (Marcolli et al., 2004). Furthermore, optical images of SOA particles are consistent with a non-crystalline state for the SOA. After collection, samples were placed in protective plastic enclosures, sealed with a vacuum food sealer, and stored in a freezer at −20 ºC until analysis (except for 24 h transit time when the samples had to be shipped to other participating laboratories on ice at 0 ºC). To prepare the sample for the viscosity measurements, some of the SOA material was scraped off the slide using a razor blade, resulting in amorphous solid particles with non-equilibrium geometries and area-equivalent diameters of ~80 to 170 µm."

*[3]* The abstract and manuscript puts a lot of emphasis on the comparison with the DeRieux et al. parameterization, showing that the observed values were "significantly higher (1–2 orders of magnitude)" than values predicted by a parameterization. I have some concerns regarding this framing. Foremost, the emphasis in the abstract distracts from the fact that this is a methods paper that has the main purpose to show that viscosity can be measured for small samples, as shown by the comparison with the TαNB bulk data. The SOM experiment nicely demonstrates that the measurement on collected samples from smog chambers is feasible. The apparent discrepancy to a parameterization is probably neither significant nor particularly important in the context of the known experimental and modeling uncertainties.

*[A3]* Thank you for the feedback, we agree with the referee's comment. To address this comment, we have put less emphasis on the comparison with the DeRieux et al. parameterization in the revised manuscript. Specific changes are listed below.

The abstract has been changed in the following manner [L19]:

"As an application to SOA, the viscosity as a function of temperature for lab-generated farnesene SOA material was measured, giving values ranging from $3.1 \times 10^6$ Pa s at 51 °C to $2.6 \times 10^4$ Pa s at 67 °C. ~~These values were significantly higher (1–2 orders of magnitude) than values predicted using a parameterization from DeRieux et al. (2018), the chemical composition of the SOA measured with a high-resolution mass spectrometer, and assuming a fragility of 10 (as done previously). These results illustrate that our new experimental approach provides important data for testing methods used for predicting the viscosities of SOA in the atmosphere~~. These results demonstrate that the viscosity as a function of temperature can be measured for lab-generated SOA material using hot-stage microscopy."

The conclusion has been changed in the following manner [L485]:

"We compared our measurements here to the predicted viscosity using HRMS measurements with the DeRieux et al. (2018) parameterization. Using the previously used value of $D_f = 10$, the parameterization under predicted our measurements by ~1–2 orders of magnitude. Considering the uncertainties in composition and uncertainties in the parameterization, this level of disagreement is not surprising. Nevertheless, this study illustrates that our new experimental approach for measuring viscosities as a function of temperature provides important data for testing methods used for predicting viscosities of SOA in the atmosphere."

*[4]* The model in this manuscript heavily relies on the assumption of D = 10 from DeRieux et al. and zero error in the Tg prediction. The pure component data underlying the D = 10 estimate has significant scatter. We actually report D = 7 for a-pinene SOA (Petters and Kasparoglu, 2020), which is an almost direct measurement. The overestimated viscosity by 1-2 orders of magnitude reported in the abstract is based on the assumed D and trusting that Tg is known perfectly. I do

not believe that better closure should be expected given the uncertainty in composition, aerosol measurement, and uncertainties in the parameterization.

*[A4] We agree with the referee's comment. We have modified the text to address the referee's comment [L448]:*

"Assuming a $D_f$ value of 10, as done previously, the predicted viscosities were ~1–2 orders of magnitude less than the measured viscosities. This level of disagreement is not surprising considering the uncertainties in the parameterization and the uncertainty in the composition of the SOA material.  For example, as shown by Angell (1997), and discussed in Gervasi et al. (2020), the $D_f$ value for organics is typically in the range of ~5–30 (Angell, 1997; Gervasi et al., 2020). Furthermore, in-source fragmentation may have occurred for farnesene SOA during the UPLC-PDA-ESI-HRMS analysis. In-source fragmentation could lead to smaller molecular weight compounds seen in the mass spectra < 250 Da and the presence of compounds containing less than 10 carbons (Fig. S6). These lower molecular weight compounds would lead to lower calculated glass transition temperatures and therefore contribute to an overall lower predicted viscosity than the experimentally determined viscosity values. Future studies are needed in order to confirm if this phenomenon is happening for farnesene SOA."

*[5]* Figure 7 explores the temperature dependence of viscosity for the SOA in the context of D, showing that no single D value can reproduce the measurements. As discussed in DeRieux et al. and further explored in Petters and Kasparoglu, Tg current viscosity data for a-pinene SOA imply a scatter in the estimated Tg of +/- 25K or with constrained D parameter and hygroscopicity +/-10K (see their Figure 3 and the associated text). Tethering Figure 7 in Kiland to a single Tg value is thus not quite appropriate. Furthermore, Tg can take on a range of values depending on the cooling rate of the substance, or in the case of the chamber the way that the SOA was formed. It would seem more logical to fit the temperature dependent viscosity data using a constant D value and extrapolate the fit to to 10^12 Pa s to find the Tg. The procedure could be repeated for a range of D values that the authors believe might be plausible. This is similar to the procedure we used in Rothfuss and Petters (2017) and Marsh et al. (2018), where we find that extrapolated Tg from a VFT fit compares to +/- 10K with literature values for select substances.  Comparing the extrapolated Tg from the measurements to the predicted Tg likely results in closure within a reasonable interval.

[A5] This is a great suggestion.  In the revised manuscript, we have followed this suggestion and added the following text [L458]:

"An alternative approach for interpreting temperature-dependent viscosity data used by Rothfuss and Petters (2017) and Marsh et al. (2018) involves fitting the VFT equation to the temperature-dependent viscosity data to estimate $D_f$, $T_0$, and $T_g$ values for the material (Marsh et al., 2018; Rothfuss and Petters, 2017).  In Fig. 8 we follow this approach.  The VFT equation was fit to the experimental data with $D_f$ and $T_0$ as free parameters, and $A$ fixed at −5 as done previously.  Using this approach, we get a $D_f$ value of 7.29 ± 0.03, a $T_0$ value of 254.4 ± 0.3 K, and a $T_g$ value of

301.7 $^{+\ 1.4}_{-\ 1.7}$ K. The $D_f$ value is very similar to the $D_f$ value of 7 reported for α-pinene SOA by Petters and Kasparoglu (2020) from temperature-dependent viscosity data (Petters and Kasparoglu, 2020). The $T_g$ value is approximately ~27 K higher than predicted with the DeRieux et al. (2018) model. However, better closure is not expected due to the uncertainties in the chemical composition of the SOA (see discuss above) and the uncertainties in $T_g$ values (± 21 K) predicted by DeRieux et al. (2018)."

We have also changed Fig. 8 to the following figure, which illustrates the new analysis:

[Figure]

**Figure 8.** Viscosity as a function of temperature for farnesene SOA material. The black data points represent the midpoint of the upper and lower limits of viscosity, measured with hot-stage microscopy (HSM). The $y$-error bars correspond to the calculated upper and lower limits of viscosity from at least two separate experiments. The temperature uncertainty ($x$-error) is a propagation of the uncertainty in the temperature control stage calibration and the largest temperature oscillation during the experiment. Note that the $x$-error bars are asymmetric because the largest temperature oscillation was always toward hotter temperatures. The black dotted line is a VFT fit to the HSM measurements, with the grey area representing the 95% prediction band of the fit. In the VFT fit, $D_f$ and $T_0$ were left as free parameters, while $A$ was fixed at −5. This fit was extrapolated to $10^{12}$ Pa s, yielding a $T_g$ of 302 K. The blue dashed line is the prediction using the method in DeRieux et al. (2018) with HRMS data, and of $D_f = 10$, which yields a $T_g$ of 275 K.

We also added the following text to the Abstract and Conclusions since we think this is an important conclusion:

"We fit the temperature-dependent data to the VFT equation and obtain a fragility parameter for the material of $7.29 \pm 0.03$, which is very similar to the fragility parameter of 7 reported for α-pinene SOA by Petters and Kasparoglu (2020)."

Thank you again for the suggestion!

[6] Regardless, given the uncertainty in aerosol chemical composition, the possibility of source fragmentation of farnesene SOA during HRMS measurement, the possibility of sample aging via partial evaporation during storage or due to chemical reactions, and the possibility that the parameterization is simply insufficient to model Tg precisely for all compositions, agreement or disagreement with the DeRieux parameterization is too unconstrained to reach all but the broadest conclusion that the data presented here are reasonable. The subtleties associated with the experiment/parameterization comparison, however, should be more thoroughly discussed in the manuscript.

[A6] We also agree with this comment. As discussed above, in the revised manuscript we have put less emphasise on the comparison with the DeRieux et al. parameterization. In addition, we have added more details on the subtleties associated with the experiment/parameterization comparison and we have added the new analysis suggested by the referee (see [A4] and [A5]).

[7] "Rothfuss and Petters used a dimer relaxation technique to measure viscosities of SOA in the range of $5 \times 10^5$ to $2 \times 10^7$ Pa s at temperatures ranging from −15 to 80 °C (Rothfuss and Petters, 2016)"

Rothfuss and Petters did not apply this technique to SOA. Application to SOA is reported in S.Petters et al. (2019) and Champion et al. (2019)

[A7] This reference has been updated. Thank you.

*[8]* "On the other hand, this means samples cannot be collected and measured at a different time and different location using this technique."

I do not understand the value of location and time in this context. The fundamental limitation of the dimer relaxation technique as described in Rothfuss and Petters is the availability of sufficient number concentration, which makes it difficult to apply the technique to aged aerosol in environmental chambers. However, absent this limitation, it would be preferable to not have to separate the particles from the gas-phase and measure it "then and there".

*[A8]* Thank you for the feedback. We have removed "this means samples cannot be collected and measured at a different time and different location using this technique". In addition, we have added the following text as suggested [L65]:

"this technique is limited to relatively high aerosol number concentrations"

*[9]* "The area and L were determined from the sequence of binary images using a MATLAB script."

The programming language is probably not important here. However, it would be helpful to state the type of image analysis that was applied as well as the potential limitations. For example, does the analysis use edge detection? Is it sensitive to lighting conditions? Does the analysis require adjustment of parameters to get it right or does it work fully automated out of the box?

*[A9]* To address the referee's comment, we have added the following discussion to the methods section [L230]:

"The image analysis software ImageJ was used to process the images recorded during the experiments. The silhouette of the needle was removed from the images, and the images were subsequently binarized. The threshold for binarization was calculated using the default ImageJ settings in all but a few cases. In the few exceptions, the default threshold method was clearly not capturing the particle's shape, so the "Mean" threshold method was used instead. For a different microscope set-up with different lighting, a different threshold method may be more accurate than the one chosen here. The roundness was then determined from the sequence of binary images using an image analysis script. This script measures the roundness of binary images by differentiating between black and white pixels, then calculating the area and $L_{Feret,max}$. The area is calculated by counting the number of black pixels. The $L_{Feret,max}$ is determined by mapping the black pixels on the perimeter of the object and subsequently finding the maximum distance between two diametrically opposed points on the convex hull that encloses the object. One limitation of this approach is that the focus and lighting during image capture can vary across experiments. However, experiments were only analyzed if a majority of the particle was in-focus during the experiments, so any uncertainty due to variability in lighting and focus is very likely to be negligible."